# Dendritic Cell Vaccines for Cancer Immunotherapy: The Role of Human Conventional Type 1 Dendritic Cells

**DOI:** 10.3390/pharmaceutics12020158

**Published:** 2020-02-15

**Authors:** João Calmeiro, Mylène A. Carrascal, Adriana Ramos Tavares, Daniel Alexandre Ferreira, Célia Gomes, Amílcar Falcão, Maria Teresa Cruz, Bruno Miguel Neves

**Affiliations:** 1Faculty of Pharmacy, University of Coimbra, 3000-548 Coimbra, Portugal; calmeiro.joao@gmail.com (J.C.); adriana_tavares.36@hotmail.com (A.R.T.); acfalcao@ff.uc.pt (A.F.); trosete@ff.uc.pt (M.T.C.); 2Center for Neuroscience and Cell Biology-CNC, University of Coimbra, 3004-504 Coimbra, Portugal; mylenecarrascal87@gmail.com; 3Tecnimede Group, 2710-089 Sintra, Portugal; 4Coimbra Institute for Clinical and Biomedical Research-iCBR, Faculty of Medicine, University of Coimbra, 3000-548 Coimbra, Portugal; dbferreira96@gmail.com (D.A.F.); cgomes@fmed.uc.pt (C.G.); 5Center for Innovation in Biomedicine and Biotechnology-CIBB, University of Coimbra, 3004-504 Coimbra, Portugal; 6Coimbra Institute for Biomedical Imaging and Translational Research-CIBIT, University of Coimbra, 3000-548 Coimbra, Portugal; 7Department of Medical Sciences and Institute of Biomedicine-iBiMED, University of Aveiro, 3810-193 Aveiro, Portugal

**Keywords:** conventional type 1 dendritic cells, CD141^+^XCR1^+^ DCs, dendritic cell-based vaccines, anti-tumor immunotherapy

## Abstract

Throughout the last decades, dendritic cell (DC)-based anti-tumor vaccines have proven to be a safe therapeutic approach, although with inconsistent clinical results. The functional limitations of ex vivo monocyte-derived dendritic cells (MoDCs) commonly used in these therapies are one of the pointed explanations for their lack of robustness. Therefore, a great effort has been made to identify DC subsets with superior features for the establishment of effective anti-tumor responses and to apply them in therapeutic approaches. Among characterized human DC subpopulations, conventional type 1 DCs (cDC1) have emerged as a highly desirable tool for empowering anti-tumor immunity. This DC subset excels in its capacity to prime antigen-specific cytotoxic T cells and to activate natural killer (NK) and natural killer T (NKT) cells, which are critical factors for an effective anti-tumor immune response. Here, we sought to revise the immunobiology of cDC1 from their ontogeny to their development, regulation and heterogeneity. We also address the role of this functionally thrilling DC subset in anti-tumor immune responses and the most recent efforts to apply it in cancer immunotherapy.

## 1. Introduction

The manipulation and education of the immune system for targeting and eliminating cancer cells has been viewed as a crucial goal of cancer therapy for decades [1,2,3]. The recent introduction of monoclonal antibodies (mAbs) blocking immune checkpoint molecules, such as programmed cell death ligand 1 (PD-L1) and cytotoxic T-lymphocyte antigen 4 (CTLA-4), in clinical practice, has been a clear success, highlighting the potential of immunotherapy in the oncology field [4,5]. Additionally, strategies directly using immune cellular effectors, such as activated natural killer (NK) cells, chimeric antigen receptors (CAR) T-cells, tumor-infiltrating lymphocytes (TILs) and tumor antigen-loaded dendritic cells (DCs), have been used to boost anti-tumor immunity, with promising results [6,7,8,9].

DCs have been clinically used for three decades, with more than 300 completed or ongoing registered clinical trials conducted to test their application for boosting anti-tumor immunity [10]. DCs are a heterogeneous population of hematopoietic cells acting on the articulation between adaptive and innate immunity [11]. They comprise several subsets with distinct phenotypical and functional capacities, distributed across the blood, skin, mucosa and lymphoid tissues. Moreover, they are proficient, displaying an unparallel capacity to acquire, process and present antigens to naïve T cells, polarizing them into effector or tolerogenic subsets [11,12,13]. Therefore, these cells orchestrate adaptive immune responses by promoting either immunity to foreign antigens or tolerance to self-molecules [14]. 

Currently, there are four approaches for exploring DCs in cancer immunotherapies: (1) non-targeted protein and nucleic acid-based vaccines; (2) antigens targeting endogenous DCs; (3) ex vivo generated DCs matured and loaded with tumor antigens; and (4) biomaterial-based platforms for the in situ recruitment and reprogramming of endogenous DCs [15,16]. Among the registered clinical trials performed with DC-based anti-tumor vaccines, the most common approach relies on the use of ex vivo differentiated DCs from leukapheresis-isolated CD14^+^ monocytes (MoDCs), cultured in the presence of granulocyte-macrophage colony-stimulating factor (GM-CSF) and interleukin 4 (IL-4) [10]. Although the gathered data shows evidence that these DC vaccines are well-tolerated and present a good safety profile, clear therapeutic outcomes are achieved in less than 15% of patients [6,10]. The common tumor-associated immune suppression in enrolled late stage patients, the tumor antigens chosen as targets and the limited functional abilities of MoDCs are some of the factors that explain this lack of efficacy [17,18]. In fact, in vitro generated MoDCs underperform in key aspects that are determinant for a successful clinical outcome, such as their ability to migrate from the injection sites towards lymph nodes and their capacity to effectively elicit strong cytotoxic T lymphocyte (CTL) responses [19,20,21,22,23,24]. As an alternative, natural circulating DCs (nDCs), despite their scarce presence in the blood, display many advantages that make them an attractive source for cancer immunotherapy.

### 1.1. What Are the Characteristics of a Robust Anti-Tumor Immune Response Elicited by DCs?

In the past two decades, the increasing knowledge on DCs and tumor biology has demonstrated that DCs protective role is highly dependent on their ability to effectively polarize CD4^+^ T cells towards the Th1 subset, to cross-present tumor antigens to CD8^+^ T cells and to both interact with and activate NK cells [15,25]. CTL-driven responses have long been recognized as central players in anti-tumor immunity and DCs have the unmatched capacity to cross-present exogenous antigens on the major histocompatibility complex (MHC)-I to naïve CD8^+^ T cells, causing their differentiation into antigen-specific CTLs [26,27]. Then, CTLs recognize antigenic peptide-MHC-I complexes presented on the surface of tumor cells, triggering their elimination [28]. Hence, DC-based vaccines must ideally present a high capacity to induce tumor-specific CTLs expressing low levels of PD-1 and CTLA-4, as well as to increase their cytolytic abilities [29,30]. It is also desirable that DC-based immunotherapies are able to enhance the expression of molecules that empower CTL migration towards tumor areas (e.g., CXCR3 and CD103/CD49a) and that promote CTL avidity for MHC-I molecules on malignant cells [31,32]. 

Efficient anti-tumor immunity also relies on the DC production of IL-12p70, which potentiates Th1 polarization and the capacity of this effector T cell population to secrete Interferon gamma (IFN-γ). Th1 lymphocytes directly eliminate tumor cells and, through the production of IL-21 and IL-2, support the differentiation and expansion of antigen-specific CTLs [33]. Moreover, CD4^+^ T cells trigger the activation and activity of tumor-infiltrating macrophages and are necessary to the adequate establishment of long-term memory CD8^+^ T cells [34,35,36]. 

Finally, it has been shown that NK cells actively participate in tumor immunosurveillance [37,38,39,40], being able to directly kill tumor cells without previous immunization or MHC restriction [41]. The cell debris resulting from NK-mediated tumor destruction fuels DCs with antigens and enhances their presentation to CD4^+^ and CD8^+^ T cells, indirectly improving anti-tumor immunity [42]. Furthermore, the interaction between NK cells and DCs is characterized by a bidirectional crosstalk that is crucial for the effector functions of each one. While activated NK cells potentiate DC maturation and kill immature DCs, thus limiting tolerogenic responses [43], mature DCs induce NK cell proliferation, cytolytic abilities, CD69 expression and the production of IFN-γ [44]. 

Therefore, the criteria that DCs should fulfil to produce a potent anti-tumor vaccine are challenging and include enhanced cross-presentation, the production of IL-12p70, the expression of high levels of co-stimulatory molecules and a strong capacity to interact with NK cells. 

### 1.2. Is There a Specific DC Subset Functionally Skilled to Achieve an Effective Anti-Tumor Immune Response?

DCs are a class of bone-marrow-derived cells present in blood, epithelial, interstitial and lymphoid tissues, originated from lympho-myeloid hematopoiesis through a series of differentiation processes [45,46]. Human DCs are represented by three major subsets that are specialized in different functions, granting diversified immune responses to different threats: CD141 (BDCA3)^+^ CLEC9A^+^ classical/myeloid DCs (cDC1), CD1c (BDCA1)^+^ classical/myeloid DCs (cDC2s) and CLEC4C (BDCA2)^+^ CD123^+^ plasmacytoid DCs (pDCs) [45,47,48,49]. 

Lymphoid organ-resident cDC1, cDC2 and pDCs were shown to possess a comparable capacity to cross-present soluble antigens, while only cDC1 and cDC2 effectively cross-present dead cell-derived material [50]. Notably, in these resident populations, cross-presentation is an intrinsic characteristic not requiring cell activation. Regarding circulating populations, several studies have demonstrated that upon activation with Toll-like receptor (TLR) ligands, such as poly:IC, cDC1 outperform cDC2 in terms of cross-presenting soluble or particulate antigens [49,50,51]. Regarding circulating pDCs, despite their lower antigen uptake capacity, when activated by TLR7 ligands, they cross-present death cell material and soluble antigens as efficiently as cDC1 [24]. Considering the ideal characteristics of DCs for eliciting a strong anti-tumor immune response referred to above, cDC1, particularly CD141^+^XCR1^+^ DCs (the homologues of murine CD8α^+^ DCs), besides being highly proficient in executing cross-presentation, produce large amounts of IL-12 and IL-15. Therefore, they excel in promoting the proper activation of NK and natural killer T (NKT) cells and priming CTLs [49,51,52,53]. Hence, strategies aiming to use CD141^+^ XCR1^+^ DCs or enhance their functions constitute a very promising and thrilling goal in the field of cancer therapy. 

In this review, we summarize the actual knowledge and most recent advances in the understanding of cDC1 biology and their clinical applicability in the context of cancer immunotherapy. 

## 2. Development, Regulation and Heterogeneity of cDC1

Ontogeny studies on murine models are beginning to unravel the development of DC subsets; however, the translation of these discoveries to human biology is not always straightforward [46]. Human DCs develop from multipotent hematopoietic stem cells (HSCs) primed by predestined-related but distinct pathways of lympho-myeloid hematopoiesis, which share a common transitory phenotype and differentiate into specific subsets by the influence of lineage-specific transcription factors, particularly IRF8 and IRF4 [45,54]. Contemporary models of lympho-myeloid hematopoiesis place the lymphoid-primed multipotent progenitors (LMPP) at the apex of all myeloid and lymphoid lineages, separated from megakaryocyte and erythroid potential (MkE) [45]. Located in the bone marrow, this precursor differentiates into the granulocyte macrophage DC progenitor (GMDP), with the potential to generate granulocyte, macrophage and DC populations [45,55]. A phenotype shift occurs when these progenitor cells start to express CD115 (M-CSFR), giving rise to macrophage DC progenitors (MDPs). Subsequently, MDPs increase the expression of CD123 and differentiate into CDPs (common DC progenitors) with the capacity to exclusively generate all three DC subsets [45,55,56]. Whereas pDCs terminally differentiate in the bone marrow, DC-restricted precursors not fully expressing the phenotype of differentiated DCs, termed pre-cDCs, migrate through the blood to lymphoid and non-lymphoid tissues, where they produce cDC1 and cDC2 (Figure 1) [56,57,58,59].

cDC1 population development is under the control of a set of key transcription factors, namely IRF8, BATF3, GATA2, PU.1, GFI1 and Id2 [45,47,56,60,61]. In accordance, a recent work demonstrated that the concomitant expression of PU.1, IRF8 and BATF3 transcription factors is sufficient for reprogramming both mouse and human fibroblasts so that they acquire a cDC1-like phenotype [62]. Other in vitro experiments have shown that FMS-like tyrosine kinase 3 ligand (Flt3L), GM-CSF, Stem cell factor (SCF), thrombopoietin (TPO), IL-6, IL-3 and IL-4 might play a role in cDC1 differentiation from human CD34^+^ progenitors [46,56,63]. Finally, Notch signaling, especially in cooperation with GM-CSF, is crucial for promoting the in vitro differentiation of cDC1 from CD34^+^ progenitors [48]; by blocking the Notch pathway at different time points, it was shown that its signal was particularly important in the beginning of the differentiation process [48]. These data point out the existence of Notch-dependent lineage bifurcation that separates pDCs and cDC1 pathways from their CDP. Notch-dependent interactions of DC precursors with stromal cells were also proposed to determine the commitment to the cDC1 lineage [48]. 

cDC1 are approximately ten times less abundant than cDC2, being present in the blood, lymph nodes, tonsils, spleen, bone marrow and non-lymphoid tissues, such as the skin, lung, intestine and liver [45,46,49]. This DC subset is characterized by a high expression of CD141, low expression of CD11b and CD11c, and the lack of CD14 and SIRPα [45,46,64]. It can also be defined by the intracellular detection of IRF8, without IRF4, which represents the standard for identifying this population [45,46]. Based on many comparative studies of human CD141^+^ DCs with their murine counterpart CD8^+^/CD103^+^ DCs, many other markers have been recognized to allow for a more accurate identification of this specific population. The cell surface expression of the C-type-lectin CLEC9A (also known as DNGR-1) and the presence of the adhesion molecule CADM1 (NECL2) and the protein BTLA (CD272) substantially increase the preciseness of the identification [45,46,49,64]. Among Toll-like receptors (TLRs), cDC1 express TLR3 and TLR9, while lacking TLR4, 5, and 7 [45,46,49,64]. Additionally, indoleamine 2,3-dioxygenase (IDO) is also highly expressed in this DC subset [45,46]. Between these receptors, human CD141^+^ DCs can also be characterized by the expression of the chemokine receptor XCR1 [45,46,47,64,65].

Following the commitment to the cDC1 pathway, the heterogeneity of CLEC9A^+^ CADM1^+^ CD141^+^ lineage in the blood was also reported, splitting this population between XCR1^−^ CXCR4^hi^ and XCR1^+^ DCs [48]. After in vitro culture of the XCR1^−^ cells, this DC subset proliferated and acquired XCR1 expression, indicating that these cells are immediate precursors of XCR1^+^ cDC1 [48]. Similarly to mice splenic cDC1, this might indicate the existence of a two-staged differentiation process of human CD141^+^ DCs: a pre-cross-presentation phase and a subsequent cross-presenting stage, in which cDC1 acquire the capacity to cross-present antigens due to the GM-CSF-mediated expression of XCR1 (Figure 1) [48,58].

## 3. The Role of cDC1 in Immunity

In light of our current knowledge on DC immunobiology, cDC1 (CD141^hi^ CLEC9A^+^ XCR1^+^) are the most effective human cross-presenting cells and thus potent CTL inducers [45,46,51]. These functional traits are empowered by the expression of molecules such as CLEC9A and XCR1. CLEC9A is a receptor for actin filaments exposed by necrotic cells, allowing their recognition, internalization and routing into the cross-presentation pathway [45,46,66,67,68]. In turn, XCR1 is the receptor of the X-C motif chemokine ligand 1 (XCL1) and is restrictively expressed by human CD141^+^ DCs [51]. XCL1, also known as lymphotactin, is selectively expressed in NK and CD8^+^ T cells at a steady-state, being enhanced during infectious and inflammatory responses [69]. Therefore, the XCL1-XCR1 axis promotes the physical engagement of NK and CD8^+^ T cells with CD141^+^XCR1^+^ DCs, which amplifies their activation state [45,46,47,51,70]. In mice, CD8^+^ T lymphocytes abundantly secrete XCL1 after in vivo antigen recognition through CD8^+^ DC presentation, increasing the number of antigen-specific CD8^+^ T cells and their capacity to secrete IFN-γ. In contrast, the absence of XCL1 was shown to impair cytotoxic responses to antigens cross-presented by CD8^+^ DCs [71]. Finally, the XCL1-XCR1 axis also plays a role in immune homeostasis, specifically in the intestine, where XCL1 produced by activated T cells attracts and enables XCR1^+^ DC maturation, which in turn provides support for T cell survival and functioning [72].

CD141^+^ XCR1^+^ DCs have been shown to be required for CD8^+^ T cell responses upon viral and bacterial infections [65]. During viral infections, pDCs accumulate at sites of CD8^+^ T cell activation, leading to the production of XCL1 by these activated T cells, which in turn attracts CD141^+^XCR1^+^ DCs. This interaction of pDCs, CD141^+^XCR1^+^ DCs and CD8^+^ T cells leads to optimal signal exchange, where type 1 interferons (IFN1) produced by pDCs improve the maturation and cross-presentation by CD141^+^XCR1^+^ DCs, thus enhancing the development of the CD8^+^ T cell response [73]. In the case of secondary infections, CD141^+^XCR1^+^ DCs are needed for the optimum expansion of memory CTLs in response to most pathogens. Moreover, the reactivation of memory CTLs relies on their interactions with CD141+XCR1^+^ DCs, an event that is triggered by the DC production of IL-12 and CXCL9 in response to NK cell-derived IFN-γ [74]. 

The relevance of cDC1 subsets in tumor immune surveillance has not yet been fully established; however, collected data from experiments with cDC1-deficient animal models, such as *Batf3*–/– mice, have revealed that these cells may play a central role [75]. In fact, the above-mentioned models fail to spontaneously reject tumor grafts and are unable to support adoptive T cell therapies or to adequately respond to an immune checkpoint blockade [76,77,78,79]. Additionally, studies specifically addressing the nature of DC subsets responsible for cross-presenting peripheral tumor antigens have evidenced migratory XCR1^+^ DC as responsible for priming CTL responses [80,81]. 

Although cDC1 are a scarce immune population in the tumor microenvironment, their abundance positively correlates with patient survival across several cancers and is an indicator of the responsiveness to therapy with immune checkpoint inhibitors [77,78,79,82]. A larger number of cDC1 were detected in sentinel lymph nodes of patients with melanoma that received combined low-dose CpG-B and GM-CSF treatment. In vivo and in vitro studies showed that these DCs were derived from blood CD141^+^ cDC1 precursors that were recruited to the sentinel lymph nodes by type I IFN and afterward maturated under the combined effect of CpG and GM-CSF. The presence of in vivo CpG/GM-CSF-induced CD141^+^DCs in sentinel lymph nodes was correlated with an increased cross-presenting capacity, T cell infiltration and patient survival [83]. Concordantly, it has been shown that regressing human tumors have higher numbers of intratumoral cDC1, which are necessary for efficient CTL-mediated tumor elimination [77].

In addition to the cross-presentation of tumor antigens to naïve CD8^+^ T cells predominantly occurring in tumor-draining lymph nodes, cDC1 also play an important role in orchestrating local anti-tumor immunity [84]. In the tumor microenvironment, cDC1s are the main source of CXCL9 and CXCL10 chemokines, which are chemoattractants for CXCR3^+^ effector cells, such as T cells, NK cells and innate lymphoid cells (ILC1) (Figure 2) [85,86,87]. By locally producing high amounts of IL-12, cDC1 help to sustain CTL cytotoxicity and INF-γ production by NK cells [53,88]. In turn, IFN-γ enhances IL-12 production by cDC1 and potentiates antigen cross-presentation [89,90]. This crosstalk assumes a higher complexity level since NK and CD8^+^ T cells produce several factors that promote the recruitment, retention and local expansion of cDC1. In addition to XCL1, NK cells produce CCL5, with mRNA levels of these two chemokines being closely correlated with gene signatures of both NK cells and cDC1 in human biopsies, and associated with overall patient survival [91,92]. This suggests that both chemokines may play an important role in attracting cDC1 from blood or surrounding tissues into tumors. Finally, NK cells were recently shown to be one of the major sources of intratumoral Flt3L. This growth factor sustains the viability and functional capacities of cDC1 within the tumor microenvironment and promotes their local differentiation from recruited precursors [93]. Although most chemokines secreted by tumor cells are chemoattractants of pro-tumorigenic immune cells, such as macrophages and Tregs, in certain circumstances, the production of CCL3, CCL4 and CCL5 mediates the recruitment of cDC1. In accordance with this, the activation of WNT/β-catenin in melanoma cells was shown to result in the ATF3-dependent repression of CCL4 transcription that in turn was correlated with decreased cDC1 numbers at the tumor site [94].

## 4. Exploiting cDC1 in Cancer Immunotherapy

Given the functional specificities of cDC1, strategies aimed at their mobilization to the tumor microenvironment, as well as their expansion and activation, are viewed as highly promising approaches for boosting anti-tumor immunity and improving the success of cancer immunotherapies (Table 1). However, the fragility and scarcity of CD141^+^XCR1^+^ DCs in peripheral blood has hindered their broader study and clinical exploitation. CD141^+^ DCs represent only ~0.03% of human PBMCs, meaning that their isolation would not result in a sufficient quantity for their use as a therapeutic vaccine [49]. Furthermore, while there are already approved clinical-grade reagents for the isolation of pDCs and cDC2, equivalent reagents are still lacking for the specific extraction of CD141^+^ DC populations. For these reasons, anti-tumor vaccines based on naturally circulating pDCs and cDC2 have already been directly assayed in several clinical trials with encouraging results (NCT01690377, NCT02993315 and NCT02692976), while there has been no registered trial for cDC1 until the present (February 2020) [95,96,97,98,99]. Although there are no clinical studies with isolated cDC1 populations, we cannot discard their possible contribution in approaches using bulk circulating DCs, such as Sipuleucel T and CMRF-56-selected blood DCs [100,101].

Nonetheless, pre-clinical studies have been carried out in order to explore the use of cDC1 in prophylactic and therapeutic vaccines. The vaccination of mice with syngeneic spleen cDC1, ex vivo-loaded with tumor lysates, caused a long-lasting effect in preventing the growth of B16-ovalbumin (OVA) melanoma cells engrafted 30 days after vaccination. Moreover, the progression of tumors in melanoma-bearing mice was halted as a consequence of the increased infiltration of CD8^+^ and CD4^+^ T cells, resulting in an 30% extent of survival [102]. The combination of this cDC1 vaccine with anti-PD-1 treatment in mice challenged with MC38 colon adenocarcinoma cells, which are partially susceptible to PD-1 inhibition, showed synergistic effects and to be more effective than monotherapy. On the other hand, when applied to an anti-PD-1 refractory B16/F10 melanoma mice model, the cDC1 vaccination reduced tumor progression, suggesting the potential benefit of this strategy in immune checkpoint refractory settings [102]. Similarly, mice vaccination with cDC1 isolated from LLC-OVA tumors, prior to the subcutaneous injection of B16-OVA cells, revealed an increased infiltration of CTLs into the tumor and a significant delay of its growth. Although it may not be effective in a tumor microenvironment dominated by immunosuppressive cell types, this strategy is still promising as a follow-up treatment where tumor-associated cDC1 may be harvested from primary tumor tissue after an operation, requiring a much lower number of cells than similar DC studies (10^4^ cells as opposed to 10^5^–3 × 10^6^ cells) [103].

To overcome the limitation imposed by the low abundance of human primary cDC1, great efforts have been made to optimize novel protocols for their ex vivo differentiation. Poulin et al. reported, for the first time, that functional CD141^+^CLEC9A^+^ DCs could be generated from in vitro expanded cord blood CD34^+^ precursors by culturing them for 12 days in RPMI supplemented with 2-mercaptoetanol, 10% heat-inactivated fetal calf serum (FCS), SCF, GM-CSF, IL-4 and Flt3L [63]. The percentage of CD141^+^CLEC9A^+^ DCs obtained was low (3–6%), but these cells were nevertheless shown to present a molecular signature characteristic of blood cDC1, such as the expression of Necl2, CD207, BATF3 and IRF8. Additionally, they respond to TLR3 and TLR8 ligands and efficiently internalize material from dead cells, cross-presenting the processed antigens to CD8^+^ T cells [63]. These protocol was optimized and the differentiated cells deeply characterized in further works [21,48,104]. The cells obtained in these enhanced conditions present phenotypical and functional features similar to those of endogenous blood CD141^+^XCR1^+^ DCs, such as the expression of CD141, XCR1, CLEC9A and the adhesion receptor CADM1, as well as the expression of several genes involved in crosstalk with NK and T cells. Moreover, XCR1^+^ CD34-derived DCs were shown to have the ability to respond to TLR3 ligands and superiorly cross-present cellular antigens when compared to their XCR1^−^ counterparts or MoDCs [21]. In another study, the culture of cord blood or G-CSF-mobilized blood CD34^+^ cells for 3 weeks in serum-free medium, supplemented with Flt3L, SCF, TPO, IL-6 and the aryl hydrocarbon receptor antagonist StemRegenin 1, was shown to produce a mixed population of functional cDC1, cDC2 and pDCs [105]. Notably, this protocol produces larger numbers of cDC1 than previous ones and given that it is based in serum-free medium, obtained cells can be easier applied in the clinic. In another cell culture setting, the repeated exposure of MoDCs to mycolic acid (MA) and/or lipoarabinomannan (LAM) was reported to promote the expression of CD141 and XCR1 within the CD1a-positive cell fraction [106]. MA- and LAM-treated MoDCs were found to secrete IL-12 in a sustained way and to acquire the ability of cross-presenting tumor antigens, triggering tumor-specific CTL responses. Finally, recent evidence indicates that the adherent fraction of peripheral blood monocytes differentiated for at least 8 days with GM-CSF and IL-4 express CD141, effectively uptake dead cells and mature when stimulated with TLR3 ligands [107]. Of note, only a variable percentage of these adherent CD141^+^ DCs express XCR1 and CLEC9A, indicating that their identity is undoubtedly distinct from endogenous cDC1.

Another strategy consists on the ex vivo differentiation of DCs from induced pluripotent stem cells (iPSC). CD141^+^ XCR1^+^ DCs have been successfully obtained from HLA-A*020(+) iPSCs, with these cells being capable of performing cross-presentation and generating antigen-specific CTL responses [108]. As they are differentiated with clinical compliant protocols, iPSCs may act as a potential source of cDC1 for application in cancer immunotherapy. Furthermore, it was recently shown that the ectopic expression of transcription factors PU.1, IRF8 and BATF3 directly reprograms mouse or human fibroblasts into cDC1-like cells [62]. These cells present a transcriptional profile close to primary splenic cDC1, increasing the expression of CD40 and CD86, as well as IL-12 secretion, upon TLR stimulation and revealing the ability to effectively capture, process and cross-present exogenous antigens to T cells. Despite the fact that the reprogramming efficiencies for human embryonic and human dermal fibroblasts were only 0.6% and 0.2%, respectively, and deeper cell characterization is still required, this work constitutes a proof-of-concept with a great clinical relevance given that dermal fibroblasts are a readily assessable starting material. 

While many obstacles still stand in the way of ex vivo manipulation of cDC1 for clinical purposes, in vivo targeting and modulation of these cells is viewed as an approach with plenty of translational potential. Most of these strategies rely on the selective expression of cell surface markers, such as CLEC9A or XCR1, enabling their specific targeting with mAbs [51,109,110,122]. In mice, the targeting of tumor antigens to cDC1, using XCL1 or mAbs as vectors, proved to elicit a potent CTL response, preventing the outgrowth of inoculated tumors without relevant adverse effects [110,111,112]. Similarly, the laser-assisted delivery of OVA fused with XCL1 to dermal XCR1^+^ DCs, in the absence of exogenous adjuvant, protected mice against OVA-expressing melanoma cells in prophylactic and therapeutic settings [113]. Furthermore, the use of mAbs as vectors has been explored to target tumor antigens to the Clec9A DC receptor in several murine cancer models [109,114,115,116,119]. This strategy has been reported to successfully increase CD8α^+^ DCs cross-presentation capacity and to enhance CTL responses, promoting tumor elimination, both in prophylactic and therapeutic contexts [114,115,119]. Importantly, targeting tumor antigens to Clec9A requires the co-administration of adjuvants, such as anti-CD40 or poly I:C, to induce effective Ag-specific CTLs; otherwise, it could result in tolerance [123]. In an attempt to circumvent the necessity of adjuvant administration, a novel delivery system was recently developed, consisting on Clec9A-targeted nanoemulsions encapsulating tumor antigens [116]. This approach was revealed to be an effective self-adjuvant delivery method since a single administration promoted antigen-specific CD4^+^ and CD8^+^ T cell proliferation and antibody responses, hampering tumor growth while increasing survival in breast cancer models [116]. 

Besides direct targeting approaches, endogenous cDC1 populations could be indirectly modulated by other cell-based immunotherapies. In accordance with this, the administration of IFNα-producing iPSC-derived proliferating myeloid cells (pMCs) to mice strongly activated host CD141^+^XCR1^+^ DCs, enhancing their CD8^+^ T cell priming capacity and thus boosting anti-tumor immune responses [120]. Furthermore, IFN-α-iPSC-pMCs administered in combination with immune checkpoint inhibitors overcame resistance to single-dose administrations, leading to longer lasting anti-tumor immunity, shedding light on the advantages that may arise from treatment combination [120]. On the other hand, allogenic T cells were recently reported to be able to act as migratory transporters, delivering antigens to resident XCR1^+^ DCs in secondary lymphoid organs [121]. Due to the strong interplay between cDC1, NK cells and effector T cells, it is plausible that TIL and NK cell-based therapies somehow influence cDC1 behavior, something that must be taken into consideration during the design of such immunotherapies.

Other approaches have been assessed in a few clinical trials, such as increasing the expression of XCL1 within the tumor in order to recruit CD141^+^ XCR1^+^ DCs. Two phase I clinical trials (NCT00062855 and NCT01713439) studied the safety and dosage of an injection of neuroblastoma cancer cells that were genetically engineered to express XCL1 and IL-2, with the first using autologous cancer cells and the second using the allogenic SJNB-JF-IL2/Lptn cell line. Both approaches aimed to enhance the immune response at the tumor site and thus lead to a more effective elimination of cancer cells in relapsed/refractory neuroblastoma [117,118]. Promising results led to a sequential ongoing phase I/II clinical trial (NCT00703222) aiming to study the safety, as well as the immunological and clinical efficacy of the combined administration of SJNB-JF-IL2/Lptn and SKNLP neuroblastoma cell lines. A fourth ongoing phase I/II study (NCT01192555) is focusing on the administration of a biological vaccine using the same XCL1- and IL-2-producing SJNB-JF-IL2/Lptn cell line in combination with the oral drug Cytoxan, with the aim of further preventing the incomplete elimination and recurrence of high-risk neuroblastoma. 

## 5. Concluding Remarks

In recent years, cDC1 have been pointed out as critical elements in the initiation of tumor-specific T cell responses, as well as in sustaining the effector activity of T and NK cells within the tumor microenvironment. However, further studies are essential for deeply understanding the biology of this DC subset in human tumors, particularly its dynamic during cancer progression and how it is conditioned by tumor immunosuppressive mechanisms. 

Despite the difficulties created by their extremely low-abundance, several approaches are being tested in order to exploit cDC1s in anti-tumor immunotherapies: ex vivo differentiation from CD34^+^ hematopoietic progenitors and from iPSCs, or the direct reprogramming of somatic cells, followed by antigen loading and administration; direct antigen targeting in endogenous cDC1; and strategies aiming to enhance their frequency and functions in tumors. Among these strategies, only cancer cells genetically engineered to express XCL1, and thus to attract cDC1, have been tested or are being currently evaluated in ongoing clinical trials [117,118]. It is expected that further approaches exploring the clinical use of these cells will strongly benefit from their combination with neoantigen targeting and the co-administration of immune checkpoint inhibitors [124,125]. Considering that blood DC populations were shown to ineffectively cross-present without adequate TLR stimulation, if these cells are used as a source, the rational selection of adjuvants or maturating agents during cell manipulation may arise as a key approach. Finally, the establishment of biomaterial-based scaffolds for the in situ recruitment and modulation of tissue resident DCs, namely cDC1 subsets, appears to be another strategy with plenty of potential for clinical translation (reviewed in [16]) (Figure 3).

The use of cDC1 in a therapeutic research context is still in the beginning stage, yet the perspective of exploiting this DC subset to improve anti-tumor immunotherapy is extremely encouraging and validation in future clinical trials is highly anticipated.

## Figures and Tables

**Figure 1 pharmaceutics-12-00158-f001:**
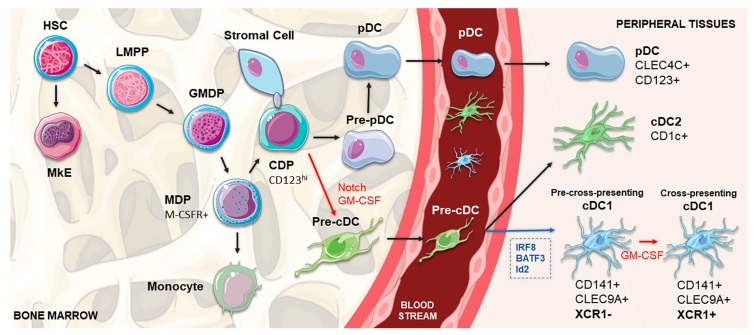
Schematic representation of cDC1 DC development. In the bone marrow, HSC gives rise to the LMPP, which settles at the apex of all myeloid and lymphoid lineages and is separated from the MkE. GMDP derives from the LMPP and produces the MDP (expressing M-CSFR), which further differentiates into the CDP, capable of generating the main DC subsets: pDCs, cDC1 and cDC2. pDCs terminally differentiate in the bone marrow, whereas pre-cDC migrate through the blood to lymphoid and non-lymphoid tissues, where they produce cDC1 and cDC2 subsets. Determining the differentiation pathway, Notch-dependent interactions of CDPs with stromal cells, in cooperation with GM-CSF, are crucial for the commitment to the cDC1 lineage, separating their pathway from pDCs. In peripheral tissues, the differentiation into CD141^+^ cDC1 is controlled by transcription factors, such as IRF8, Batf3 and Id2 (shown in blue). After engagement with the CD141^+^ pathway, the existence of a two-staged differentiation process of cDC1 populations is proposed, in which XCR1-negative cDC1s under the influence of GM-CSF (shown in red) acquire the expression of XCR1, representing the shift between a pre-cross-presentation phase and a subsequential cross-presenting stage. cDC, conventional dendritic cell; CDP, common DC precursor; DC, dendritic cell; GM-CSF, granulocyte-macrophage colony-stimulating factor; GMDP, granulocyte macrophage DC progenitor; HSC, hematopoietic stem cell; LMPP, lymphoid-primed multipotent progenitor; M-CSFR, macrophage colony-stimulating factor receptor; MDP, macrophage DC precursor; MkE, megakaryocyte and erythroid potential; pDC, plasmacytoid dendritic cell.

**Figure 2 pharmaceutics-12-00158-f002:**
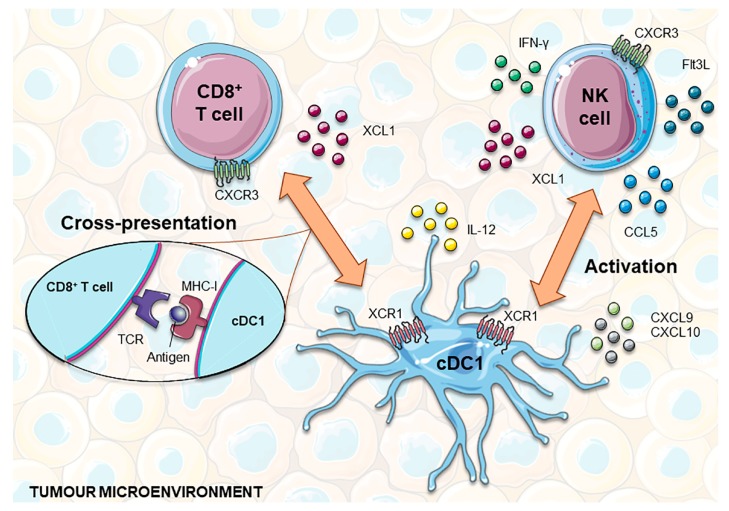
cDC1 interplay with CD8+ T and NK cells to develop anti-tumor responses. NK and CD8^+^ T cells express XCL1, which attracts XCR1^+^ cDC1 into the tumor microenvironment. In addition, NK cells can also produce CCL5, helping to recruit this subset of DCs. In turn, cDC1 are the main source of the chemokines CXCL9 and CXCL10, chemoattractants for T and NK cells. Functionally, cDC1 are highly capable of cross-presenting tumor antigens via MHC-I to CD8^+^ T cells and producing IL-12, which promotes T cell cytotoxicity and the production of INF-γ by NK cells. Furthermore, NK cells produce Flt3L that holds up the viability and functional capacities of cDC1 within the tumor microenvironment and can also promote their local differentiation from recruited precursors. cDC1, classical dendritic cell 1; Flt3L, FMS-like tyrosine kinase 3 ligand; IFN-γ, interferon gamma; MHC-I, major histocompatibility complex I; NK, natural killer; TCR, T cell receptor; XCL1, X-C Motif Chemokine Ligand 1.

**Figure 3 pharmaceutics-12-00158-f003:**
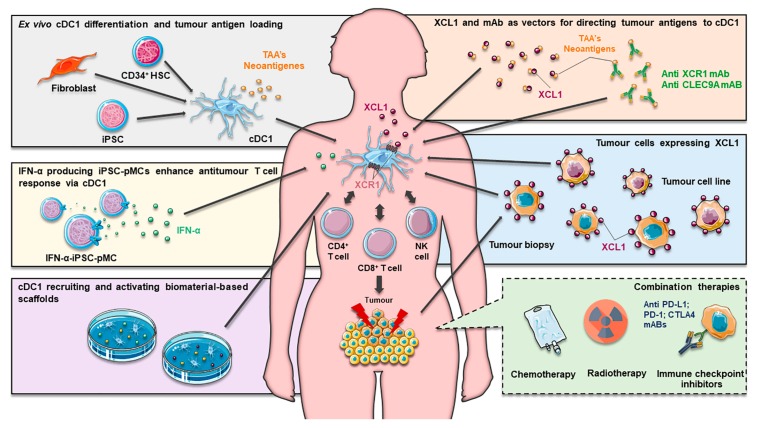
Summary of current approaches exploring cDC1 in immunotherapy. To overcome the scarcity of natural circulating XCR1^+^ DCs, which hampers their clinical application, new protocols have allowed their ex vivo generation from CD34^+^ precursors and iPSCs and their reprograming from fibroblasts. As the resulting cells excel at the cross-presentation and generation of CTL responses, they allow for the formulation of innovative DC-based vaccines, where antigen loading with different TAAs and promising neoantigens potentially generate superior outcomes. Endogenous cDC1 can also be in vivo targeted by using carriers such as XCL1 or monoclonal antibodies to deliver antigens in a specific manner. Likewise, autologous and allogeneic tumor cells can be genetically engineered to express XCL1 and target XCR1^+^ DCs. Furthermore, IFN-α-iPSC-pMCs can potentiate XCR1^+^ DC activity by releasing IFN-α in vivo. Additionally, biomaterial-based scaffolds can be designed to release factors that recruit and activate DCs. These strategies aim to boost antigen-specific CD8^+^ T clone proliferation and responses, as well as enhance CD4^+^ T and NK cell activity. All these approaches can be potentially combined with conventional cancer therapeutics, namely radiotherapy and chemotherapy, but also with more recently developed immunotherapy strategies, such as immune checkpoint inhibitors. cDC1, Classical dendritic cell 1; CTL, cytotoxic T lymphocyte; DCs, dendritic cells; IFN-α, interferon alpha; IFN-α-iPSC-pMCs, IFN-α producing induced pluripotent stem-cell-derived proliferating myeloid cells; iPSCs, induced pluripotent stem-cells; NK, natural killer; TAAs, tumor-associated antigens; XCL1, X-C Motif Chemokine Ligand 1.

**Table 1 pharmaceutics-12-00158-t001:** Overview of existing approaches aiming to explore cDC1s for anti-tumor immunotherapies.

Approach	Studied Species	Cell Subset	Differentiation Cocktail	Antigen Type	Target/Tumor Model	Combination Therapy	Ref
ex vivo differentiation	Human	CD34^+^-derived CD141^+^ CLEC9A^+^ DCs	SCF, GM-CSF, IL-4 and Flt3L	-	-	-	[21,48,63,104]
	Human	CD34^+^-derived cDC1	Flt3L, SCF, TPO, IL-6 and StemRegenin1	-	-	-	[105]
	Human	Monocyte-derived CD141^+^ XCR1^+^ DCs ^1^	GM-CSF and IL-4	-	-	-	[107]
	Human	CD141^+^ XCR1^+^ DCs	MA and LAM	-	-	-	[106]
	Human	iPSC-derived CD141^+^ XCR1^+^ DCs	GM-CSF, SCF, VEGF and BMP4	Melan A	^2^		[108]
	Human and murine	Fibroblast-derived cDC1	PU.1, IRF8 and BATF3	-	-	-	[62]
Naturally occurring cDC1	Murine	Natural cDC1	-	UV-irradiated tumor cell lysates	B16 melanoma MC38 colon adenocarcinoma	Anti-PD-1	[102]
	Murine	Tumor-derived cDC1	-		B16 melanoma LLC lung carcinoma		[103]
mAb- or XCL1-based direct in vivo targeting	Murine	CD8α^+^ DC	-	IgG2a mAb Ovalbumin	-	-	[109]
	Murine ^3^	XCR1^+^ DC	-	Ovalbumin	EL4 thymoma	-	[110]
	Murine	CD8α^+^ DC	-	Ovalbumin	P3X63Ag8.653 myeloma	-	[111]
	Murine	CD8α^+^ DC	-	Ovalbumin	B16 melanoma	-	[112,113]
	Murine	CD8α^+^ DC	-	Ovalbumin	B16 melanoma and lung pseudometastases	-	[114]
	Murine	CD8α^+^ DC	-	MUC1	MC38 colon adenocarcinoma	-	[115]
	Murine	CD8α^+^ DC	-	Nanoemulsion Ovalbumine	PyMT-mChOVA breast cancer and lung metastases B16 melanoma HPV-related TC1 cancer	-	[116]
	Human ^4^	Allogeneic neuroblastoma cells	-	-	Neuroblastoma	-	NCT01713439 NCT00703222 [117]
	Human ^4^	Autologous neuroblastoma cells	-	-	Neuroblastoma	-	NCT00062855 [118]
	Human ^4^	Allogeneic neuroblastoma cells	-	-	Neuroblastoma	Cytoxan	NCT01192555
WH-based direct in vivo targeting	Murine	CD8α^+^ DC	-	Ovalbumin	B16 melanoma	-	[119]
Indirect in vivo targeting	Murine	IFN- α -iPSC-pMCs			B16 melanoma EL4 thymoma MC38 colon adenocarcinoma CT26 colorectal adenocarcinoma 4T1 breast cancer	Anti-PD-1/anti-PD-L1	[120]
	Murine	CD8α^+^ DC	-	Allogeneic T cells	-	-	[121]

^1^ Adherent fraction; ^2^ Functionality assays; ^3^ Transgenic mice expressing human XCR1; ^4^ Clinical trial.

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
