# Peer review of "Dendritic Cell Vaccines for Cancer Immunotherapy: The Role of Human Conventional Type 1 Dendritic Cells"

_pharmaceutics, 2020, doi:10.3390/pharmaceutics12020158_

Round 1

Reviewer 1 Report

This is an excellent, well written and detailed review of the potential application of CDC1 (CD141+ mDC) based therapy for cancer. I feel this work will be of great interest to the readership. I have a few minor suggestions below:

The rationale behind progressing to blood-derived DC from Mo-DC could be discussed. eg. Limited results from current Mo-DC trials (eg ADAPT trial) and the low migratory potential of injected MoDC vaccines (Verdjik. Cancer Res., 15 (2009), pp. 2531-2540) Line 101-113. When discussing blood DC sources of vaccines, a brief discussion of pDC (Jonada De Vries group’s work eg Tel et al. 2012Cancer Res; 73(3); 1063-75) and CD1c DC: including the possibility for them to be immunosuppressive (Kassianos et al Eur J Immunol. 2012 Jun;42(6):1512-22.) and the fact that there have been limited results from clinical trials (eg. Prue et al J Immunother. 2015 Feb-Mar;38(2):71-6) The development and ontogeny section is excellent Lines 185-190. Review articles are referenced alongside the major statement that CD141 DC are the most efficient cross-presenting DC. It may be better to reference the major original research papers describing this as it is perhaps the most important feature of this dendritic cell subset. Key references here are the Jongbloed paper (ref 48), Bachem paper (ref 50) and perhaps Tel Blood. 2013;121(3):459-467, which suggests that pDC may have a comparible cross-presenting capacity depending on activation stimuli. Section 5. This section omits any mention of bulk DC preparations that contain an enrichment of CD141+ DC. For eg. Sipuleucel T and the CMRF56 selected blood DC vaccine (Fromm et al. 2016 May 5;5(6):e1168555) Section 5. A discussion of the logistics of direct extraction of CD141 DC leukapheresis that would be beneficial. For example, the PBMC yield, and the resulting number of CD 141 DC extracted (could use Jongbloed et al. ref 48 as a guide, Miltenyi also have a MACS kit).

Author Response

On behalf of all co-authors, we would like to express our gratitude for the comprehensive, constructive and complete revision, which contributed to improve the quality and focus of the manuscript. We also appreciated very much the stimulating words given by the Referees along the revision.

The changes made on the manuscript were attentive to all those considerations and we feel that the end product is now of greater interest for the readers of the Pharmaceutics Journal.

Minor grammar and spelling changes are evidenced by “track changes” and sections of text introduced to answer Reviewers questions were highlighted (yellow).

Reviewer 1

Referee comment:

“The rationale behind progressing to blood-derived DC from Mo-DC could be discussed. eg. Limited results from current Mo-DC trials (eg ADAPT trial) and the low migratory potential of injected MoDC vaccines (Verdjik. Cancer Res., 15 (2009), pp. 2531-2540) Line 101-113.”

Our comments:

As requested by the Reviewer we added several sentences justifying the rational for the progression of blood-derived DCs from the initially used Mo-DCs. The suggested references were added and explored. In the revised version of the manuscript the added sentences are in the section 2 (Introduction).

Referee comment:

“When discussing blood DC sources of vaccines, a brief discussion of pDC (Jonada De Vries group’s work eg Tel et al. 2012Cancer Res; 73(3); 1063-75) and CD1c DC: including the possibility for them to be immunosuppressive (Kassianos et al Eur J Immunol. 2012 Jun;42(6):1512-22.) and the fact that there have been limited results from clinical trials (eg. Prue et al J Immunother. 2015 Feb-Mar;38(2):71-6).”

Our comments:

As requested by the Reviewer we added a brief discussion about pDC and cDC2, adding the suggested references. In the revised version of the manuscript the added sentences are in the section 2 (Introduction).

Referee comment:

“The development and ontogeny section is excellent Lines 185-190. Review articles are referenced alongside the major statement that CD141 DC are the most efficient cross- presenting DC. It may be better to reference the major original research papers describing this as it is perhaps the most important feature of this dendritic cell subset. Key references here are the Jongbloed paper (ref 48), Bachem paper (ref 50) and perhaps Tel Blood. 2013;121(3):459-467, which suggests that pDC may have a comparable cross-presenting capacity depending on activation stimuli.”

Our comments:

As suggested, key references describing CD141+ cross-presentation capacity were maintained in the manuscript and reviews references were deleted (Ref 72). The suggested original research on pDC cross-presentation comparable activity was added.

Referee comment:

“Section 5. This section omits any mention of bulk DC preparations that contain an enrichment of CD141+ DC. For eg. Sipuleucel T and the CMRF56 selected blood DC vaccine (Fromm et al. 2016 May 5;5(6):e1168555)”

Our comments:

As required, we added the information regarding bulk preparations that may contain CD141+ DC.

Referee comment:

“Section 5. A discussion of the logistics of direct extraction of CD141 DC leukapheresis that would be beneficial. For example, the PBMC yield, and the resulting number of CD 141 DC extracted (could use Jongbloed et al. ref 48 as a guide, Miltenyi also have a MACS kit).”

Our comments:

As required, we added the information about the isolation of CD141+ DCs, however Miltenyi has only available an isolation kit for research and not for clinical applications for now.

Reviewer 2 Report

The manuscript provides a comprehensive overview of cDC1 biology as well as the properties that make them a promising tool for anticancer immunotherapy. Moreover, in the absence of clinical trials able to substantiate the role of cDC1 in DC-based immunotherapies, authors summarise preclinical studies encouraging cDC1 clinical use, as well as the efforts made either to target them in vivo or differentiate them ex vivo. 

Comment:

Clinical application of cancer immunotherapies based on cDC1 represents a novel therapeutic approach that exploits naturally circulating DC subsets, instead of monocyte-derived DCs (moDCs). Until present, clinical trials exploring the role of other DC subsets such as BDCA1+ (CD1c) DCs or pDCs have been performed, producing encouraging results. In order to support the role of cDC1 in DC-based immunotherapies and to draw firm confirmation, comparision to moDCs as well as other circulating DC subsets will be needed. For these reasons and considering the title proposed by the authors for the present manuscript, "Next-generation Dendritic Cell Vaccines for Cancer Immunotherapy: The Role of Human Conventional Type 1 Dendritic Cells", before focusing on cDC1, authors should provide an overview of DC vaccination approach based on circulating DCs and results obtained so far. Alternatively, if the authors prefer to maintain the present manuscript structure, my suggestion is to make some changes to the title (e.g. Role of cDC1 in antitumor immunity: future approach in immunotherapy.)

The manuscript is well referenced but, concerning cross-presentation mechanism, there is a relevant study (Segura et al, 2013 Similar antigen cross-presentation capacity and phagocytic functions in all freshly isolated human lymphoid
organ–resident dendritic cells, JEM), not cited in the text, showing similar cross-presentation capability of lymphoid organ dendritic cells indicating that all resident DC subsets are potential targets when aiming for cross-presentation and therefore for the design of new vaccination strategies. In addition, the same study reveals that blood DCs do not cross-present efficiently without TLR-mediated activation, step most likely bypassed in lymphoid organs. Altogether these findings raise the question wether other source of DCs, such as tissues, could be taken into account for clinical use. Please discuss these points.

Author Response

On behalf of all co-authors, we would like to express our gratitude for the comprehensive, constructive and complete revision, which contributed to improve the quality and focus of the manuscript. We also appreciated very much the stimulating words given by the Referees along the revision.

The changes made on the manuscript were attentive to all those considerations and we feel that the end product is now of greater interest for the readers of the Pharmaceutics Journal.

Minor grammar and spelling changes are evidenced by “track changes” and sections of text introduced to answer Reviewers questions were highlighted (yellow).

Reviewer 2

Referee comment:

“Clinical application of cancer immunotherapies based on cDC1 represents a novel therapeutic approach that exploits naturally circulating DC subsets, instead of monocyte-derived DCs (moDCs). Until present, clinical trials exploring the role of other DC subsets such as BDCA1+ (CD1c) DCs or pDCs have been performed, producing encouraging results. In order to support the role of cDC1 in DC-based immunotherapies and to draw firm confirmation, comparison to moDCs as well as other circulating DC subsets will be needed. For these reasons and considering the title proposed by the authors for the present manuscript, “Next-generation Dendritic Cell Vaccines for Cancer Immunotherapy: The Role of Human Conventional Type 1 Dendritic Cells”;, before focusing on cDC1, authors should provide an overview of DC vaccination approach based on circulating DCs and results obtained so far. Alternatively, if the authors prefer to maintain the present manuscript structure, my suggestion is to make some changes to the title (e.g. Role of cDC1 in antitumor immunity: future approach in immunotherapy.)”

Our comments:

We maintained the present manuscript structure and, as suggested by the reviewer, changed the title to: “Dendritic Cell Vaccines for Cancer Immunotherapy: The role of Human Conventional Type 1 Dendritic Cells”. As suggested, we also added a brief overview of the clinical trials with naturally circulating DCs (pDCs and cDC2).

Referee comment:

“The manuscript is well referenced but, concerning cross-presentation mechanism, there is a relevant study (Segura et al, 2013) Similar antigen cross-presentation capacity and phagocytic functions in all freshly isolated human lymphoid organ–resident dendritic cells, JEM), not cited in the text, showing similar cross-presentation capability of lymphoid organ dendritic cells indicating that all resident DC subsets are potential targets when aiming for cross-presentation and therefore for the design of new vaccination strategies.”

Our comments:

As suggested, cross-presentation capacities of lymphoid-resident and circulating DCs subsets was discussed (section 2) and Elodie Segura paper was referenced.

Referee comment:

“In addition, the same study reveals that blood DCs do not cross-present efficiently without TLR-mediated activation, step most likely bypassed in lymphoid organs. Altogether these findings raise the question whether other source of DCs, such as tissues, could be taken into account for clinical use. Please discuss these points.”

Our comments:

As required by the Reviewer, in the revised form of the manuscript, this point is discussed in section 2. We add also a paragraph in “Concluding Remarks” discussing the possibility of exploring endogenous tissue resident DCs as clinical strategy.